# Energy-Efficient Industrial Internet of Things Software-Defined Network by Means of the Peano Fractal

**DOI:** 10.3390/s20102855

**Published:** 2020-05-18

**Authors:** Jesus Jaime Moreno Escobar, Oswaldo Morales Matamoros, Ixchel Lina Reyes, Ricardo Tejeida-Padilla, Liliana Chanona Hernández, Juan Pablo Francisco Posadas Durán

**Affiliations:** 1Escuela Superior de Ingeniería Mecánica y Eléctrica, Instituto Politécnico Nacional, Mexico City 07340, Mexico; oswmm2001@yahoo.com (O.M.M.); megamikurayami6@gmail.com (I.L.R.); lchanona@gmail.com (L.C.H.); jpposadas@gmail.com (J.P.F.P.D.); 2Escuela Superior de Turismo, Instituto Politécnico Nacional, Mexico City 07630, Mexico; ricardotp75@hotmail.com

**Keywords:** energy-efficient systems, industrial Internet of things, software-defined network, peano fractal curve, swarm intelligence

## Abstract

The Industrial Internet of Things (IIoT) network generates great economic benefits in processes, system installation, maintenance, reliability, scalability, and interoperability. Wireless sensor networks (WSNs) allow the IIoT network to collect, process, and share data of different parameters among Industrial IoT sense Node (IISN). ESP8266 are IISNs connected to the Internet by means of a hub to share their information. In this article, a light-diffusion algorithm in WSN to connect all the IISNs is designed, based on the Peano fractal and swarm intelligence, i.e., without using a hub, simply sharing parameters with two adjacent IINSs, assuming that any IISN knows the parameters of the rest of these devices, even if they are not adjacent. We simulated the performance of our algorithm and compared it with other state-of-the-art protocols, finding that our proposal generates a longer lifetime of the IIoT network when few IISNs were connected. Thus, there is a saving-energy of approximately 5% but with 64 nodes there is a saving of more than 20%, because the IIoT network can grow in a $3^{n}$ way and the proposed topology does not impact in a linear way but $\log_{3}$, which balances energy consumption throughout the IIoT network.

## 1. Introduction

Advances in technology at different times have given rise to three industrial revolutions, communication systems, intelligent robots, and the Internet of Things (IoT), which are thought to lead humanity to the fourth-industrial revolution by connecting devices, people, data, and processes. IoT is a new generation of networks made up of several elements for the identification, perception, communication, computing, services, and semantics of the information obtained from the environment, allowing connectivity between the digital and the physical world using different technologies [1,2].

In 2020, IoT is expected to provide a huge amount of intelligence available in the cloud to billions of mobile devices, delivering an enormous amount of new values with more than 55 million applications available to almost any human. This implies an interconnection of four million people over the world [3]. This has had a strong impact on the environment that surrounds people both in households (IoT) and industry (Industrial Internet of Things (IIoT)). The latter combines autonomous and intelligent machines with advanced predictive analytics as well as gets information from humans and machines collaboration, yielding improvements in productivity, efficiency, and reliability, which all together give rise to a communication infrastructure that allows each device to be accessible in a barrier-free context without sacrificing integrity and information security [4].

One fundamental technology for IIoT performance is Wireless Sensor Networks (WSNs) as they permit collecting, processing, and sharing data of different natures along the whole network. WSNs have been used in applications such as healthcare, monitoring of the environment or traffic, and industry, among others [5]. In IIoT, on the one hand, connected devices are restricted in terms of power supply, training factor, external wiring, computing power, bandwidth, memory, and storage. Thus, it is necessary to determine the points where the sensors are placed to avoid lack of connectivity among devices [6].

On the other hand, in a plant distribution, where an IIoT network is located with a WiFi access point (CN), it is possible to connect all devices to the Internet or an intranet as the hub node. Therefore, if an IIoT Sensor Node (IISN) seeks to share parameters with another one, they must do so by connecting to the CN first.

Therefore, this article proposes an algorithm that effectively allows the transmission and distribution of parameters in all devices connected to WSN. This makes it necessary to use many networked sensors to obtain information in real time and, consequently, an efficient IIoT network, which in turn allows the protocol proposed in this work to be applied in an industrial plant. The present article is organized as follows. Section 2 shows different reliable communication protocols when operating WSNs for efficient energy consumption. Section 3 describes the axiom, production rules, and parameters proposed for building the Peano fractal curve. Section 4 specifies the proposal, based on swarm intelligence. In Section 5, we verify our algorithm by comparing its performance with respect to RRDLA, DETR, and REER protocols. In Section 6, we present the conclusions of this work. Finnally, Appendix A contains a list of abbreviations and notations to facilitate the reading of this work.

## 2. Related Work

Wireless Sensor Networks (WSNs) are a series of electronic chips working collectively to gather, process, and analyze information from several environments, allowing the construction of a global network of intelligent devices. In this global network, data are sensed and shared among its sensor nodes, generating and sending information to be analyzed by a centralized or distribution system.

The use of many embedded devices to process and communicate information produces many data to be transmitted and stored, which can give rise to high costs. Moreover, the achievement of an optimal-network performance depends on the following aspects: the placement of nodes, its efficiency on handling many data, the interoperability between devices with different technologies, appropriate identification for each device, security and protection of devices, confidence in data collected, energy consumption in devices, and efficient use of radio frequency spectrum between devices making up the network or from other systems [7]. Therefore, different works have considered several approaches to efficient-energy consumption during data transmission in WSNs.

Li et al. [8] proposed the reliable and energy-efficient routing (REER) protocol based on some centralized and distributed power control techniques for guaranteed routing of WSNs, addressing the cases of a single emission and many emissions in the presence of unreliable links, and taking into account link error rate as a function of transmission rate. They found that routing of an emission used less power because it had unreliable communication links, while power used during many emissions was very close to optimal, concluding that the second alternative was better for achieving energy savings during data transmission throughout WSNs. However, they did not consider how interference caused by transmission from other sensor nodes affects power consumption within the WSNs.

Liu et al. [9] developed the delay-energy tradeoff with reliable communication (DETR) protocol to make energy consumption more efficient by applying industrial control in WSNs, guaranteeing quality, real-time delivery and reliability in data transmission. In DETR, a node first calculated its power consumption, delay, and reliability using local information generated by its neighboring nodes, to chose which node would receive its data packet. The candidate node to take the next jump would be the one with the highest level of residual energy within a set of nodes. As for real-time delivery, the node with the highest speed would be chosen to take the next jump. In addition, to achieve greater reliability, the route having the best reliability would be selected, instead of discarding packets. The simulation results show that the DETR generated the best performances, compared to the SPEED and AVR protocols, in terms of energy consumption, network life, and transmission delay.

Amarnath and Sujatha [10] proposed the use of WSNs as part of an expert monitoring system for power generation. Through solar and wind energy, this type of smart grid is an efficient means of electricity transmission at low cost and high security. In addition, an efficient energy-saving mechanism monitored with a minimum of energy was used and it displayed results in the web browser.

Bajrami and Murturi [11] developed a system with both mobile and web browser applications to monitor environmental conditions. The data collected were stored in the cloud where they were analyzed to have insight of the prevailing conditions. This system was tested in a laboratory as well as telecomm server for getting data that indicate the reliability of their proposal and its successful application for monitoring humidity and air temperature in both places. This proposal had the possibility of warning the user of any change in configured value ranges through email that could bring the possibility of making changes due to the conditions and keeping a history of its behavior for future analysis.

Mohamed et al. [12] carried out a study on network applications for WSN and routing protocols that were energy efficient. Network applications were categorized according to complications in designing protocols for homogeneous proactive networks. The authors suggested that power overload and route selection were the most effective aspects to consider for achieve useful life and efficiency of the network. According to them, for routing protocols, the main factors that affect the energy efficiency of the network are the following: (i) long periods of operation of the sensor; (ii) limited energy of the sensors; (iii) environmental factors; (iv) nature of the environment whose variables are going to be measured; (v) flow of information from many devices to where data are centralized; and (vi) random deployment of sensors usually used in many applications. This resulted in fault tolerance and overload on network during configuration and reset, and route selection for data transmission.

Mostafaei and Menth [13] proposed the SD-WSNs configuration protocol for data transmission by applying the software-defined networking (SDN) in WSNs. This protocol was designed to be used for energy efficiency, routing, mobility, QoS, and reliability topics. In that algorithm, the centralized controller generated energy savings in each sensor node because: (a) each sensor node was in charge of complying with QoS; (b) sensor nodes collaborated with each other for routing decisions; (c) sensor nodes needed to send broadcast messages to their neighbors to find them and estimate the active time of each sensor node; (d) there were the same security threats anywhere on the network; (e) interaction among the sensor nodes determined where they should move; (f) the sensor nodes must interact with each other to better location and higher transfer power; (g) all sensor nodes may fail to direct traffic; and (h) interaction among all sensor nodes made it efficient to manage the WSN. Therefore, the development and application of routing configuration protocols in WSNs is encouraging to contribute to energy savings in this type of network, by avoiding redundant communication among the sensor nodes through intelligent operation.

Javadi et al. [14] developed the learning automaton based topology control (LBLATC) protocol to configure routing in WSNs from the idea that a lower range to transmit data made the most of the resources of each sensor node and, consequently, reduced its energy consumption. In that algorithm, the learning automaton of each sensor node allowed one to choose the lowest-transmission range based on the signal strength generated by the learning automaton of its neighboring sensor nodes. By simulating the LBLATC performance, an increase in the performance of the WSNs of 15% compared to other protocols was obtained, making possible to save energy and extend the useful life of these networks, since as few links as possible were used during data transmission. However, they did not consider the mobility of the sensor nodes within the network, since this property of the network topology allows the sensor nodes to move from one region to another within the same WSN.

Mostafaei [15] developed the reliable routing distributed learning automaton (RRDLA) algorithm to streamline the performance of the network, and therefore reduce energy consumption within it, by means of the modeling and simulation about the way parameters were connected taking into account the following constraints: end-to-end delay, packet delivery rate, network lifetime, and the number of times the data were transmitted. the smallest number of nodes required with dynamic and highly reliable links to achieve the optimal route that would allow routing in the WSN, during the transfer of information, to comply with the Quality of Service (QoS) restrictions in the delivery of end-to-end data, was found. By simulating the results, Mostafaei concluded that his algorithm performed better, in terms of less delay in end-to-end data delivery, a longer network lifetime by occupying fewer nodes during data transmission and energy saving because RRDLA requires a fewer data transmissions. Thus, it is necessary to take into account learning automaton and multiconstrained QoS routing to generate routing configuration protocols for WSNs.

According to the above, advances in software for WSN can be classified into nine categories: (i) energy efficiency, including a reclassification in terms of lifetime, coverage control, and node clustering; (ii) routing; (iii) mobility to locate nodes; (iv) reliability of data transmission outside the network; (v) guaranteed level of service quality; (vi) network management in terms of configuration, provisioning, and maintenance; (vii) location of each one of nodes for several applications of the WSNs; (viii) wireless power; and (ix) security of each of devices on the network.

Due to the expansion of working frequency band, demand for the development of wide-band and ultra-wide-band, radio engineering systems have expanded to increase rate of information transmission, immunity to noise, and information capacity of all kinds of radio engineering systems.

Radio engineering fractal geometry of a transceiver device or any electronic acoustic, optoelectronic and radio information system appear to be appropriate for solving challenging problems in electronics of traditional radio [16].

For instance, because the Peano fractal curve has infinite dimension, this curve tends to occupy all the two-dimensional space that contains it, which is why Li et al. [17] proposed a Peano fractal antenna for Ultra-High Frequency (UHF) Partial Discharge (PD) online monitoring of transformer with small size and multiband. They established a rough formula for calculating the first resonant frequency of their Peano fractal antenna. The simulations and experimental results obtained were compared with those generated by the Hilbert fractal antenna, yielding the following: (i) external dimension of the Peano fractal antenna was smaller than Hilbert fractal antenna when their performances were similar; (ii) Peano fractal antenna could receive electromagnetic waves from the front of the antenna; and (iii) PD signals measured by the Peano UHF fractal antenna were slightly wider than the signals detected by the Hilbert antenna, implying that Peano fractal antenna was slightly more suitable for pattern recognition when analyzing waveforms of detected UHF PD signals.

Likewise, Potapov [16] explained the design of broadband or multiband fractal antennas and other basic electrodynamic structures for innovative information and communication technologies, based on the fact that the study of the interaction of pre-fractal geometries with electromagnetic fields had allowed discovering extremely complex electrodynamic structure properties. This researcher’s findings on the behavior of small pre-fractal antennas have led to useful design criteria for optimal small-wire antennas, and this knowledge can be applied to the use of pre-fractal geometries in other modern electrodynamic devices. In addition, Potapov demonstrated the advantages of using multilayer fractals in resonators, filters, and other radio and electrodynamic structures.

Therefore, it can be said that the introduction of fractals can contribute significantly to the future of radio electronics, since all previous and contemporary radio electronics are exclusively based on the theory of integer functions. Similarly, this suggests the advantage of space-filling fractal topologies in a very efficient way, i.e., these topologies allow optimizing the transmission of data in WNSs within smaller spaces and, therefore, make energy consumption more efficient within an IIoT network. To the expansion of the working frequency band, the demand for the development of wide-band and ultra-wide-band radio, engineering systems has expanded to increase the rate of information transmission, immunity to noise, and information capacity of all kinds of radio engineering systems.

Based on the above, to achieve a most efficient energy consumption when WSNs are transmitting data in a WSN, we design an adaptive reconfiguration algorithm for linking all sensor nodes of an IIoT using the Peano fractal topology to be a filling-space curve, in order to avoid the use of centralized and distributed protocols for a reliable routing of these networks, since distributed control of WSNs by itself generates lots of communication overhead among the sensor nodes. Therefore, in our proposal, we do not use hub nodes but the sensor nodes or microcontrollers (Industrial IoT sense Nodes (IISN)) of the IIoT network share all the parameters only by linking in a dynamic way with their adjacent sensor nodes. With the proposed algorithm, it is possible to have access to the entire IIoT network in real time, allowing to reconfigure the topology of the IIoT network when a new node is added.

## 3. Peano Fractal Curve

Mandelbrot [18] called curves with fractal dimension greater than their topological Dimension 1, whose structure always remains when they expand or contract and that do not have a tangent at any point, fractal curves, such as the Peano and Hilbert curves tending to fill the space containing them. In 1890, Peano proposed the first fractal curve taking the numbering base, not the decimal, but Base 3. The Peano curve is continuous, with a not differentiable function at any point and is of Dimension 1 filling a square of Dimension 2 [19]. However, Peano did not give any graphic idea of how this curve could be, since his work was strictly analytical. In 1891, Hilbert published a version in which he did give a geometric image of a curve that filled the square but using Base 2 instead of Piano’s Base 3 [20].

After Hilbert’s work, the construction of Peano could be resumed and something similar might be done on a recurring basis. A unit or initial square is divided into nine equal squares that are numbered 1–9. By means of segments, the centers of the squares are joined according to numbering order, that is, beginning with the one that occupies the position at the bottom left and ending with the one in the upper right position, always using adjacent squares. The next step is to divide each of squares from the previous division again into nine other squares, so there are yielded 81 squares numbered 1–81. Next, the centers of the squares are joined according to their order numeric, so that the polygon curve does not cut itself, as shown in Step 2 of Figure 1. In this way, the squares of each division are recurrently subdivided into nine squares and a line is drawn that joins the centers without self-intersections, i.e, without cutting itself.

Lindenmayer formally described a plant growth using a parallel rewriting system known as L-system [21], which is used, due to its simplicity and recursion, to build fractal curves. In a L-system, there are (re)production (or rewriting) rules that indicate how the growth of the system or phenomenon under study should be, as well as the initial state of the system or phenomenon called the axiom. Therefore, L-systems are string rewriting machines whose production rules apply simultaneously to all symbols in the input string. The chains generated by L-systems must be complemented with a graphic interpretation of them to visualize the generated fractals. Thus, Papert developed a very simple graphical interface to visualize the symbol strings, which is based on graphs traced by an imaginary turtle moving in a plane [22]. This turtle commands it in the form of symbols used in L-system, which are just common letters of the alphabet and some special symbols such as ⥀ or ⥁.

From Lindemayer’s L-systems and Pepert’s graphic interpretation, in this work, the production rules are formulated to construct the Peano fractal curve. Below are the first set of instructions for the turtle: Φ (move forward one step in length l and draw a line from the old to the new position), ⥀ (turn left, counterclockwise, in an angle ψ), and ⥁ (turn clockwise, to an angle ψ).

Therefore, the production rules for first stage of the Peano curve construction start with only one-line segment as the axiom Φ. This curve (which is later replaced by simple line-segments) is coded as follows. After a step forward, the turtle turns left or right, or it can go straight. If it turns right, it must turn left twice with line-segments between three turns. Thus, the turtle has managed to reach a point where the curve meets itself, and there is a choice: to continue in the same direction or to turn left. In both cases, the turtle will trace the upper loop of the generator counterclockwise or clockwise; and only then the last-line segment will have finished at the end of the curve.

In terms of turtle commands, these two alternatives can be described as follows (in any case, ψ=90 is chosen):Φ⥁Φ⥀Φ⥀Φ⥀Φ⥁Φ⥁Φ⥁Φ⥀Φ, andΦ⥁Φ⥀Φ⥀ΦΦ⥀Φ⥀Φ⥀ΦΦ.

If the turtle continues in the same direction at the point of the first decision, we have:ΦΦ⥁Φ⥁Φ⥁ΦΦ⥁Φ⥁Φ⥀Φ, orΦΦ⥀Φ⥀Φ⥀ΦΦ⥀Φ⥀Φ⥁Φ.

The complete L-system can work with any of these decisions, for example, it can be established:
Axiom:ΦProduction rules:Φ⟶ΦΦ⥀Φ⥀Φ⥀ΦΦ⥀Φ⥀Φ⥁Φ
⥀⟶⥀
⥁⟶⥁Parameter:ψ=90

## 4. Energy Efficient IIoT Algorithm

### 4.1. Proposal Overview

Wang et al. [23] and Han et al. [24] mentioned that most of sensor nodes involved in IIoT have static hierarchical routing, in order to avoid complicated routing which takes a long time to complete. These authors suggested that any IIoT architecture should have the following three layers (Figure 2).

Control layer: In this layer, there are Control Nodes (CN) managing the Gateway Nodes (GN). Its objective is to collect data from the Sense nodes (SN) and send them to a cloud server, in order to obtain important information for higher layers. Thus, these nodes designate the GN based on their features.Gateway layer: This layer has mostly devices capable of high-level processing because in this layer a routing protocol needs to be established, tending sometimes to be complex since they also manage the state of the sensors located in the lower layer. The GN forwards data measured by the SNs, routing them to the cloud through a CN for some other entity to process or interpret. The GN also calculate the times taken by the SN to share their data. Finally, there are no constraints for communication between GN.Sense Layer: In an industrial area, information is collected by sensors located in a nearby perimeter area. SNs measure data, detect important environment variables, and send the information to GN. The SNs are divided into two large groups, namely Trigger-Based Sense Nodes (TBSN) and Periodic Sense Nodes (PSN), whose categorization is based on how they collect the data and their transmission-frequency. Thus, TBSNs send their data throughout the network when there is an event which exceeds certain threshold or for a particular event. If this threshold is not exceeded, this type of sensor is kept waiting. Otherwise, the PSN sends data with a frequency measured in regular time intervals and regulated by the GN. Both TBSN and PSN measure data and store them in an internal memory, and then transmit them from their communication ports.

To suggest how sensor nodes could behave in a WSN, we used the work from Beni and Wang [25], since they were the first to introduce the term Swarm Intelligence (SI) and use it on their cellular robot system. Therefore, we consider an Artificial Intelligence (AI) tool to be an SI algorithm. Then, IS is an AI technique for studying the collective behavior to regulate non-centralized systems. In a basic SI system, agents interact with each other and with their environment autonomously. Each of these agents or sensor nodes is governed by simple rules and not by a hub sensor node. Despite that, they adequately solve their interactions or common actions, working out as a single entity with its own intelligence.

The five greatest benefits of swarm intelligence are:Derivative-free optimizationRobustnessFlexibilityEasy administration and implementationLow computational and economic costs

Thus, schemes such as the one shown in Figure 2 making up a three-layer system and whose energy saving lies in the balance of traffic loads, avoiding point-to-point communication between SNs, are always communicating by means of a GN. On the contrary, we propose to merge the top two layers, i.e., form the control gateway layer and maintain the sense layer, as shown in Figure 3.

With a two-layer system, our control link layer will control routing by adapting network topology from software, thus the gateways will decide who replicates the data obtained from SNs. That is why our energy saving proposal is not only to balance the traffic load avoiding point-to-point communications among sensors, but also the dynamic reconfiguration of the network.

### 4.2. System Model

The SI efficiently uses the entire swarm, optimizing itself, and thus providing smaller algorithms in conception but jointly rebuffed. This causes flexibility and low computational cost for solving complex problems to avoid them growing without a central operation. We define the proposed system model using the Labeled Transition System (LTS) described by Moller and Struth ([26] Chapter 11). In this way, our system is defined as a triple $\gamma =\left(\sigma ,\alpha ,\delta \right)$, as follows:A set of σ statesA set of α events or actions.A set of δ⊆σ⟶α⟶σ transitions from one state to another, where δ denotes a transition relation

Therefore, the simplified system model is defined by Equation (Equation 1),
(1)$$HubNode{⟶}^{IISN_{i}}ListOfNodes{⟶}^{LNR_{i},IISN_{i}}FractalSorting{⟶}^{Reconfiguration}HubNode$$

The proposed system, in terms of a LTS, is expressed as follows: In the HubNode state, we may estimate $IISN_{i}$ for evolving into the ListOfNodes state, changing to FractalSorting state by means of estimating $LRN_{i}$ and replicating $IISN_{i}$. The current state will be kept until there is a new control node, which will invoke reconfiguration action in whole the IIOT network.

In Figure 4, the main IISN or hub node is chosen as the closest one with the maximum bandwidth to the $CN_{i}$. Then, all the nodes broadcast for each CN to find its own List of Reachable Node ($LNR_{i}$). With this information, the bandwidth is estimated in Mbps and a new topology is calculated for an industrial plant. In this topology, when a SN is added, the algorithm is capable of configuring again the entire topology. With the new topology, each $IISN_{i}$, considered as a new node in the IIoT network, is added depending on the position it keeps in the path of Peano fractal sorting. Then, it is measured how efficient the new topology is when sharing parameters in the network. Finally, both the WiFi Peer-to-Peer (P2P) and SoftAP algorithms are reconfigured.

### 4.3. First Stage: Hub Sensing Entities

The present algorithm is considered as the CN, which is a WiFi node placed in a random position within the industrial plant and which has at least one SN.

First, all the IISNs will try to connect to the CN. If a communication channel is not established with it, each one enables WiFi direct mode. Thus, the IISNs successfully establish data transmission and reception inactivate in WiFi direct mode, waiting for instructions from the CN. It is the latter that performs a broadcast mode, thereby managing to flood the network with packets whose objective is to measure bandwidth with each member or SN of the IIoT network.

Once broadcast is finished, an identification called IISN Network Identifier (IINI) is added, containing the type of IISN connected, e.g. IIoT-ESP8266 or IIoT-RPi3, as well as its identification of 1 byte-length inside the IIoT network. Therefore, IRU = IoT-ESP8266-i indicates a connected ESP8266 microcontroller, whose identification is understood as the *i*th GN system with connection regardless of what type of sensing variables it can acquire.

Only some IISN establish connection to the CN due to obstacles for the signal such as furniture or walls. According to Equation (Equation 2), the connected IISNs must not exceed $3^{k}$ in total.
(2)$$IIoTe=\sum_{i=1}^{3^{2k}}IISN_{i},$$
where IIoTe is the number of embedded systems or GNs connected in the network, *i* is the index of the IINI, and *k* is the iteration or level of the Peano fractal,
(3)$$n=\left\lceil \log_{9}\left(IIoTe\right)\right\rceil .$$

Hence, it is possible to define $IISN_{i=1}$ as the closest IISN to CN with an effective connection, since it gets the best connection in terms of bandwidth. Thereby, the first sensor node of the IIoT network is indexed as $IISN_{1}$.

### 4.4. Second Stage: Identification of $LRN_{i}\left(t\right)$

Once the main node of the IIoT network has been identified, $IISN_{i=1}$, as well as the rest of the embedded systems, $IISN_{i\;neq1}$, the WiFi direct mode is enabled and a table is filled, containing the bandwidth of each of these GN, and then a List of Reachable Nodes (LRN) is generated.

Later, each network configuration can generate up to $9^{k}$ LRNs. The proposal is to function as the first layers since each $IISN_{i}$ behaves as a CN at the same time being a GN for the neighboring SNs. Hence, each $LRN_{i}$ contains the bandwidth of all $IISN_{i}$ that must be connected to at least one other $IISN_{i}$.

The LRNs are shared throughout the network and the $\sum_{i=1}^{9^{k}}IISN_{i}$ SNs know each of the LRNs and, thus, also all the interconnection topology of the IIoT network. This step is where the swarm intelligence takes place, since all the SNs work individually to improve the IIoT network performance.

### 4.5. Third Stage: Sorting by Means of the Peano Fractal Production Rules

Based on the above, each node has an initial index in the network, which at first is not optimized. To optimize it, all the elements must be numbered, according to the Peano fractal construction and the position where each sensor node passes.

Linear indexing is developed to identify the array $IISN_{i}$ as a vector. The array of IISN microcontrollers is defined as P and the resulting vector interspersed as $\vec{P}$, with $3^{k}\times 3^{k}$ as the size of P and $9^{k}$ as the size of $\vec{P}$, where *k* is the the Peano curve level (or iteration).

Algorithm 1 generates a Peano’s mapping matrix, θ, with a level *k*, expressing each curve as nine consecutive indices. The *k*-level of θ is acquired by concatenating four different transformations θ into the previous level n−1. In the Peano’s mapping matrix, $\bar{\beta }$ refers to a 90-degree rotation of β and $\beta^{T}$ is the β (or $\bar{\beta }$) linear algebraic transposition.

Figure 5 shows an example of the mapping matrix θ at the level k=2. Therefore, every $IISN_{i}$ in $P_{\left(i,j\right)}$ is saved and ordered in ${\vec{P}}_{\theta_{\left(i,j\right)}}$, where $\theta_{\left(i,j\right)}$ is the location index of $IISN_{i}$ in $\vec{P}$. Figure 5b shows the position within a room and the MRI of each $IISN_{i}$, that is, the best way to communicate for $IISN_{i}$ is the $IISN_{i+1}$, increasing the bandwidth. Algorithm 1 was designed to give index and spatial position to each sensor node or device connected to the network at a given moment within an industrial warehouse. That warehouse was used to find the best reconfiguration, according to the number of IIoT devices found, be they microcontrollers, video surveillance cameras, or smartphones.
**Algorithm 1:** Function to generate the $3^{k}\times 3^{k}$ size Peano theta mapping matrix.
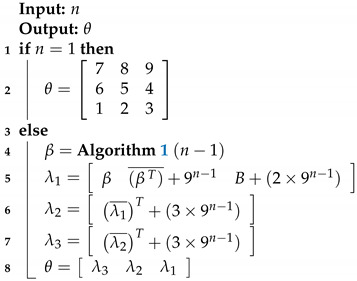


According to Algorithm 1, sensor nodes *i* are consecutive and communicated to the sensor node i+1. In the current topology, there is no $IISN_{i}$ in a certain position *i*; the index that is the closest i−1 is reached; and this $IISN_{i}$ is considered as a Non-Significant Node (NSN). Otherwise, the sensor node is considered a Significant Node (NS). When this node $IISN_{i+1}$ is NSN, the node $IISN_{i}$ reaches a NS, increasing *i* until reaching it.

To know the significance of the IIoT network, we propose a Network Significance String (NSS) with $9^{k}$ bits, 0 for NSN and 1 for NS. The NSS is propagated to $\sum_{i=1}^{9^{k}}IISN_{i}$ within a few nanoseconds. Each $IISN_{i}$ measures or assesses certain parameters such as temperature, humidity, $CO_{2}$, and adjustment of the ambient lights. To differentiate the parameters, we propose a 1-byte marker that specifically indicates what parameter a sensor measures, or Sensor Identifier (SId), in units of degrees Celsius or volts. For example, Table 1 shows some measurable parameters by a $IISN_{i}$, as well as a 1-byte SId to point out up to 256 parameters.

Each parameter measured by a $IISN_{i}$ is represented by SId Marker (SIdM), which is a 2-byte identifier and is divided into three parts: sign, exponent, ε, and mantissa, μ. The most significant bit of SIdM is the sign of the parameter: 0 for positive and 1 for negative. The last ten significant bits are used for the location of μ, which is defined as:(4)$$\mu =\left\lfloor 2^{10}\left(\frac{Parameter}{2^{R-\varepsilon }}-1\right)+\frac{1}{2}\right\rfloor .$$

Equation (Equation 5) expresses how ε is obtained, which is stored in the remaining five bits of SIdM,
(5)$$\varepsilon =R-\left\lceil \log_{2}\left|Parameter\right|\right\rceil ,$$
where *R* is the number of bits used to represent the highest value of a certain parameter, defined as
(6)$$R=\left\lceil \log_{2}\left[\max\left\{Parameter\right\}\right]\right\rceil \;.$$

## 5. Performance Evaluation

### 5.1. Experiment Setup

IWe designed an industrial plant distribution and monitored the following amount of IISN:Forty-six IIoT-ESP8266: Low-cost IEEE 802.11 b/g/n Wi-Fi chip working with the TCP/IP protocol, 32-bit RISC CPU, 64 KB of RAM for instructions and 96 KB of RAM for data.Eighteen IIoT-RPi3: CPU + GPU: Broadcom BCM2837B0 1.4 Ghz, Bluetooth 4.2, Dual Band IEEE 802.11 b/g/n Wi-Fi 2.4/5 Ghz, 300 Mbps Gigabit Ethernet network card.

We used a CN to connect the IIoT network to the Internet as the initial node with the following characteristics: Tenda-AC1200 wireless router, wireless signal repeater, IEEE of the 802.11 b/g/n RJ45 standard with 4 ports configured at 300 Mbps. The IISN distribution in an industrial plant using a second-level Peano fractal curve depicted by Figure 6 was used. This figure shows both the ideal configuration in a pictorial representation (Figure 6a) and the actual configuration achieved after applying the proposed algorithm (Figure 6b). The connections shown in Figure 6b were joined by straight lines to show the topological approximation in a drawing after applying the proposed algorithm.

### 5.2. Quality of Service and Impact of Node Density

The performance of this proposal was evaluated by estimating the Node Density (ND) impact, in order to measure the Quality of Service (QoS). Then, the proposed algorithm performance was estimated in terms of Packet Delivery ratio (PDR) and the End-to-End Delay (E2ED). This performance was compared with similar protocols:Reliable Routing with Distributed Learning Automaton (RRDLA) [15]: The RRDLA is composed of four steps:(a)Initialization: Network of learning automaton is formed in a distributed way, namely each IoT device supports one learning automaton. In addition, the learning automaton of entire action plan or topology is calculated.(b)Acquisition: Selection of a scarce nodes’ number with high packet delivery ratio is carried out.(c)Broadcasting: Transmitting information by means of selected nodes is performed.(d)Forwarding: Undelivered information is transmitted to the sink node.That algorithm presents both iteration and delay thresholds, thereby flooding the network with hello messages and putting all the nodes on listening for these messages. Then, it randomly chooses a learning automaton, to which a consecutive index is assigned. Hence, by using a QoS rule and network requirements, it generates a topology. Since certain nodes are disabled, this algorithm saves energy by using residuals from inactive nodes.Delay–Energy Tradeoff in wireless sensor networks with Reliable communication (DETR) [9]: This algorithm creates $F_{1}$, which is the set of devices inside the radio range and sends a broadcast message to all nodes. Then, that algorithm calculates information delay time at the same time it calculates its power consumption. Thus, by means of probabilities, it calculates what node has the maximum tradeoff energy rate, thus it forwards this information to the entire network. Hence, the DETR algorithm can be expressed as follows:Define $F_{1}$.Broadcast *HELLO* message.For each j IoT-device, $j\epsilon F_{i}$.Estimate the delay time of each package.Calculate Energy Consumption for each node.IF this device obtains the best delay-energy tradeoff, choose it as the main device.ELSE Choose it as the sink device.End.Reliable and Energy-Efficient Routing (REER) [8]: This algorithm assumes that certain nodes have a certain preset power, thus it generates multi-point communication through multicast. This algorithm bases its savings on energy consumption when choosing the most optimal linear route, comparing the best available options. Other features that can be highlighted are the following:(a)A multicast routing protocol is performed with the purpose of energy consumption saving(b)Two times of the minimum in a one-to-one communication mode(c)Estimation of optimal or near optimal power assignments and communication routes(d)Minimization of expected total energy(e)Decide communication routingUnicast.Multipath.Multicast.

Hence, all these algorithms, including ours, send messages to know the existing number of nodes. However, our proposal is based on fractal-production rules, offering the best interconnection route to form a topology which does not apply any thresholds or comparisons until finding the best option recursively.

The same network parameters, proposed by Zorzi and Rao [27] and Karp and Kung [28], were employed to make a real comparison to the state-of-the-art algorithms.

To understand the experiments carried out, first we define E2ED as the time it takes for a packet to reach the receiver from the moment it is emitted until it is received, being then the average sum of the differences between the delays sending and receiving [29]. PDR is defined as the probability of recovery from the original packets by receiver before their lifetime expires [30]. From these definitions, we have Equation (Equation 7) defining the rate (κ) of change between the lowest and the highest delay as follows:(7)$$\kappa \left(\tau \right)=\frac{E2ED{\left(\tau \right)}_{\max}-E2ED{\left(\tau \right)}_{\min}}{PDR{\left(\tau \right)}_{\max}-PDR{\left(\tau \right)}_{\min}},$$
where κ→0 when E2ED is constant, but κ→∞ when PDR is constant, regardless E2ED. In addition, reliability requirements is an important term in QoS, since it is necessary to measure the impact of node density or τ. The same experimental design was taken from Mostafaei [15] and we compared this proposal with the algorithms developed by Niu et al. [31] and Zeng et al. [32].

The comparison between E2ED and PDR was made in two ways. The first, in terms of reliability requirements and node density, is shown in Figure 7a,b, respectively. The lower is the PDR, the better is the performance, while the E2ED works otherwise, i.e., when it increases the performance of the algorithm, it also increases this metric. On the one hand, Figure 7a shows a comparison of our proposal in terms of the reliability requirements. When PDR is increased in REER or DETR, they tend to increase their E2ED, while the RRDLA algorithms and our proposal have a lower E2ED, regardless of the number of packets that are delivered, that is both algorithms keep their recovery reliability on the IIoT network.

On the other hand, Figure 7b shows a comparison of the present proposal in terms of the node density. When PDR is increased in REER or DETR, they tend to significantly increase their E2ED, while the RRDLA and the present proposal have a low E2ED regardless of the number of sensor nodes connected to the network. Thereby, both RRDLA and the present proposal reduce the delay and increase the reliability of delivering the packets adequately, regardless of the sensor nodes connected to the IIoT network. Therefore, the best results are obtained by our proposal anr the RRDLA algorithm. This happens because both protocols minimize the delay for data transfer from the node sensors when the packet receiver retrieves it.

In summary, using all the information in Figure 7, the present proposal is slightly better than the other algorithms, exceeding them by at least 2%. Hence, propagation delay in the IIoT network is highlighted. Both the present proposal and RRDLA remain stable regardless of the node density or QoS. Other algorithms such as DETR and REER obtain much larger delays than this proposal.

The node density impact and quality of service in κ(τ) are shown in Table 2. The best result obtained by this proposed is highlighted in **bold**, while the second best result is obtained by RDDLA algorithm, in *italics*.

Finally, when a sensor node shares data in the IIoT network, it takes less time to reach the receiver when the proposed algorithm is used. Hence, the recovery propagation of the original packet before its lifetime expires is significantly increased.

### 5.3. Impact of Node as Expression of Energy Consumption

This section presents a comparison between the energy consumption and the increase in SNs, as well as how much energy saving is achieved when using the proposed methodology. To calculate the amount of kilowatts per hour (kWh) in a continuous-operation day, the following formula was applied
(8)$$kWh=\frac{V\times I\times 24}{1000},$$
where V = 3.3 V, I = 70 mA for the IIoT-ESP8266, while for IIoT-RPi3 I = 350 mA.

Figure 8 shows the distribution in an industrial warehouse without walls or obstacles for all CNs to establish connections to a centralized router. According to Figure 9a, the proposed algorithm presents greater energy savings as more SNs are added to the IIoT network, compared to a centralized topology. In the centralized or star topology, SNs were placed in a distance where all sensor nodes had the CN connection, while the proposed topology disperses the SNs throughout the industrial plant (Figure 6a).

Therefore, the energy saving was 0.002 kWh with two SNs but it increased to 0.143 kWh with 64 SNs. This is expressed as a percentage of energy savings in Figure 9b. Thereby, with a few nodes there was a saving of approximately 5% but with 64 nodes there was a saving of more than 20%. The growth in energy savings is because the IIoT network can grow in a $3^{n}$ way, and the proposed topology does not impact in a linear way but $\log_{3}$, which balances energy consumption throughout the IIoT network, as if this topology understood where there is a need for more or less energy and balances it.

Since the RRDLA [15], DETR [9], and REER [8] algorithms carried out simulations using the network simulator software ns-2 [33], these works did not report their energy consumption data, thus it is only possible to compare these algorithms with the present proposal in terms of reliability requirements or node density (as done in Section 5.2), and it is impossible to compare them in terms of energy efficiency as they are logical and not physical topologies. The present algorithm was simulated with the ns-2 software but was also physically configured in the room shown in Figure 6a.

## 6. Conclusions

Reducing energy consumption is a primary objective in the design of any communication system for IIoT networks. Most of this energy can be saved by adding more sensor nodes or wireless sensor nodes to these networks, since most of the information obtained is redundant due to the sensor nodes geographic location. Therefore, the efficient diffusion scheme considers parameters distribution already proposed. However, this continues to imply high communication costs, making it difficult to share the chosen parameters. That is why in this work a light diffusion algorithm in wireless sensor networks has been proposed. From the analysis carried out, we confirm that the reconfiguration scheme of a IIoT network generates a longer lifetime of the network as more efficient information on the parameter to be sensed, with respect to the traditional schemes or topologies.

When a few sensor nodes are connected, there is a saving of approximately 5% but with 64 sensor nodes there is a saving of more than 20%. The growth in energy savings is because the IIoT network can grow in a $3^{n}$ way, and the proposed topology does not impact in a linear way but $\log_{3}$, which balances energy consumption throughout the IIoT network, as if this topology understood where there is a need for more or less energy and balances it.

## Figures and Tables

**Figure 1 sensors-20-02855-f001:**
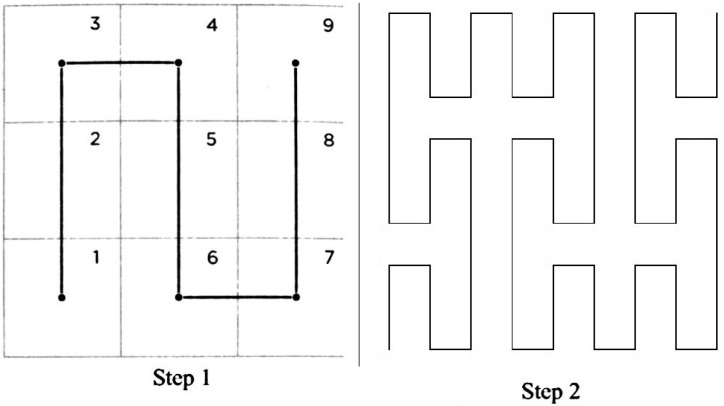
Construction of the Peano curve: At each step, a line segment is replaced by nine segments reduced by a factor of three.

**Figure 2 sensors-20-02855-f002:**
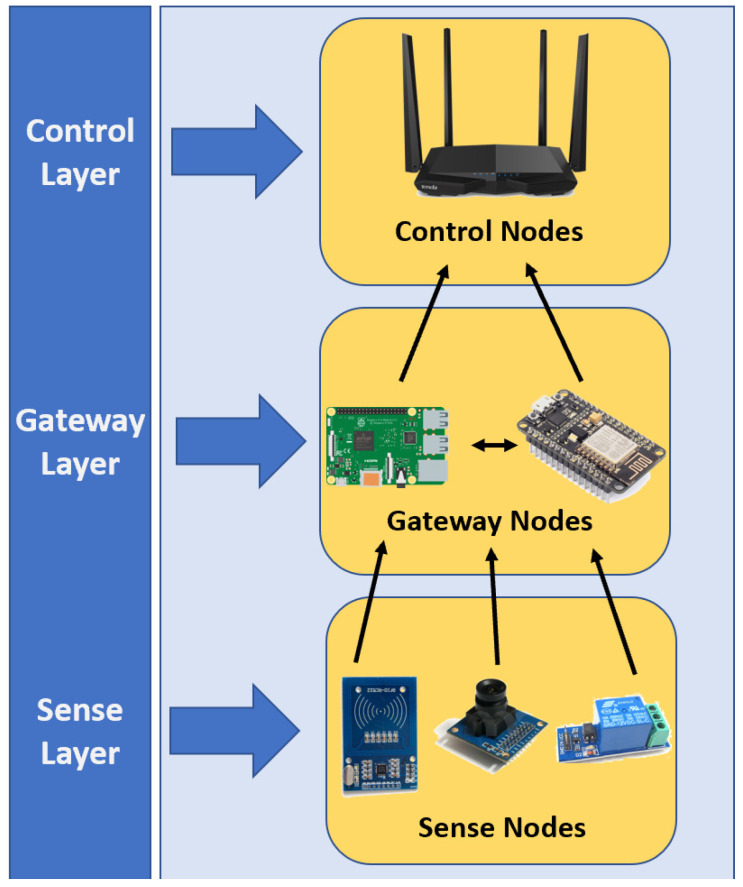
Architecture into the sensing entities domain in three layers.

**Figure 3 sensors-20-02855-f003:**
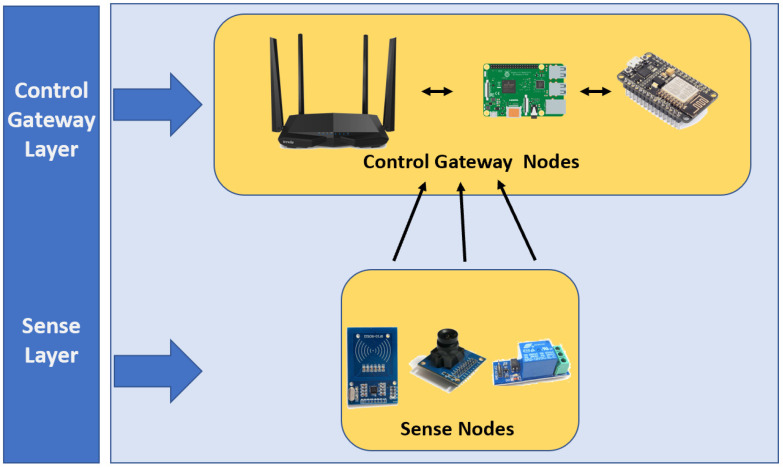
Proposed architecture placing the sensing entities domain in two layers.

**Figure 4 sensors-20-02855-f004:**
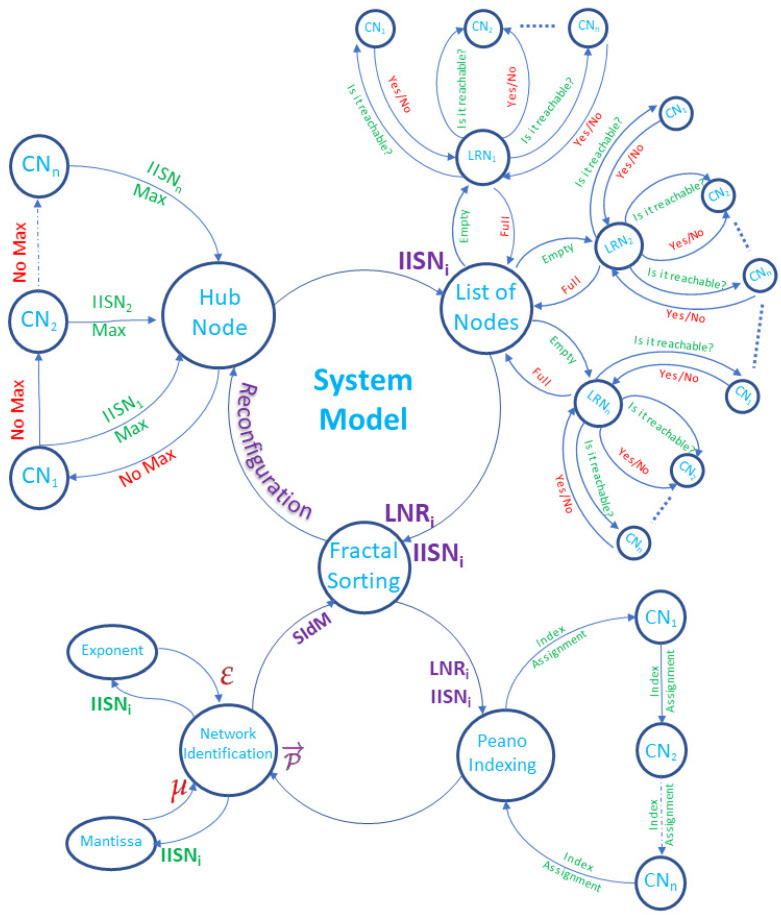
Proposed System Model using a labeled transition system.

**Figure 5 sensors-20-02855-f005:**
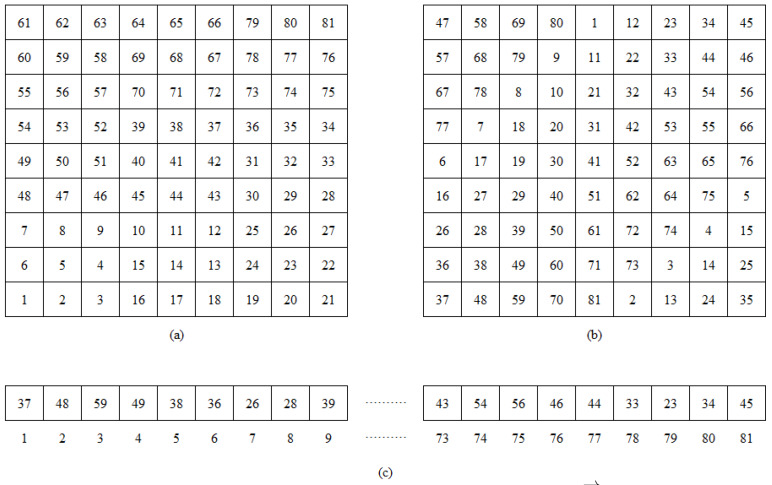
(**a**) Matrix θ; (**b**) matrix P; and (**c**) vector $\vec{P}$.

**Figure 6 sensors-20-02855-f006:**
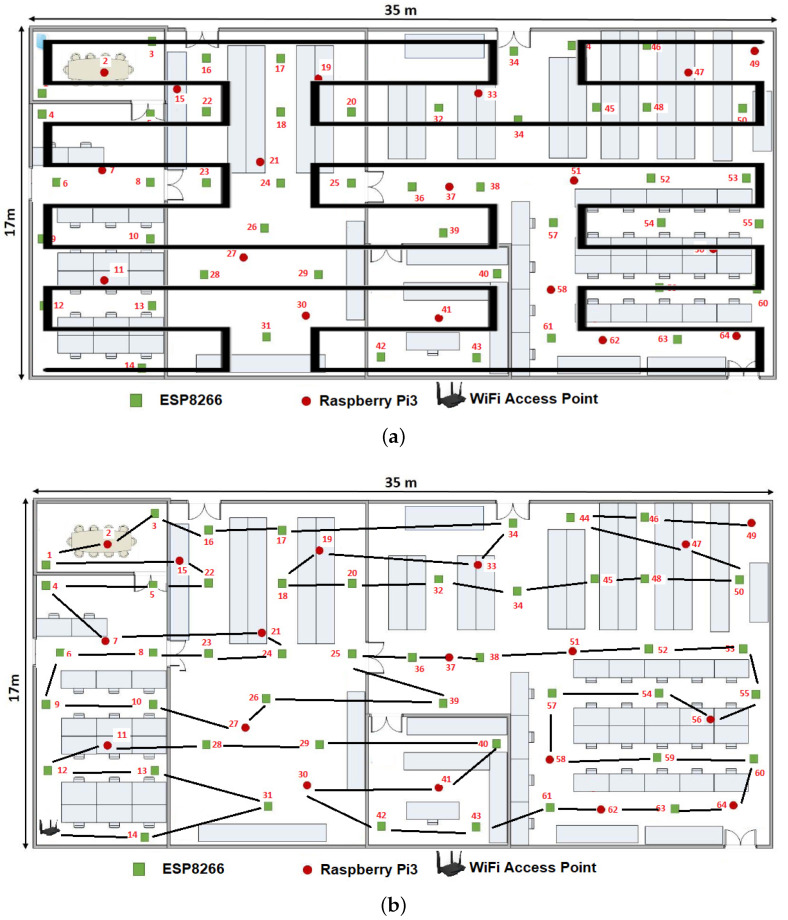
IISN distribution in an Industrial Plant using the Peano fractal curve: (**a**) ideal connection representation; and (**b**) actual connection representation.

**Figure 7 sensors-20-02855-f007:**
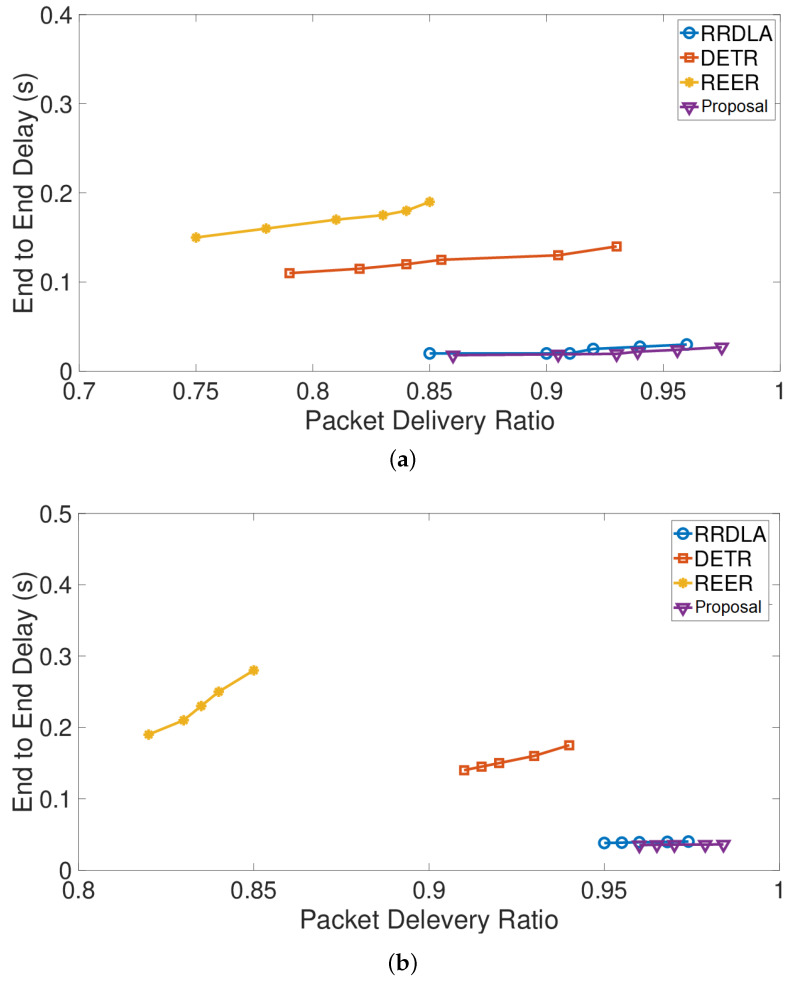
QoS and ND impact: (**a**) reliability requirements; and (**b**) node density.

**Figure 8 sensors-20-02855-f008:**
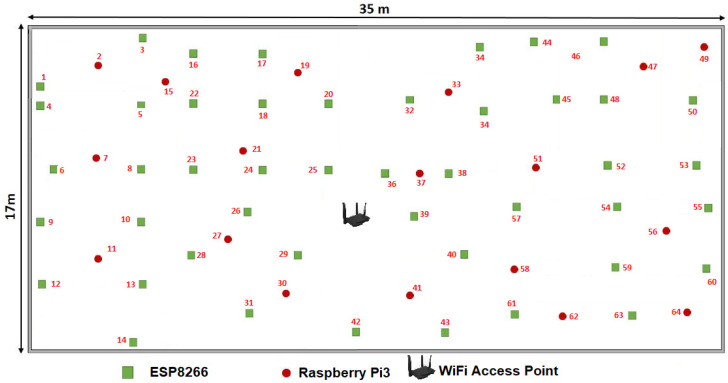
IISN distribution in an industrial plant to connect all sensor nodes in a centralized topology.

**Figure 9 sensors-20-02855-f009:**
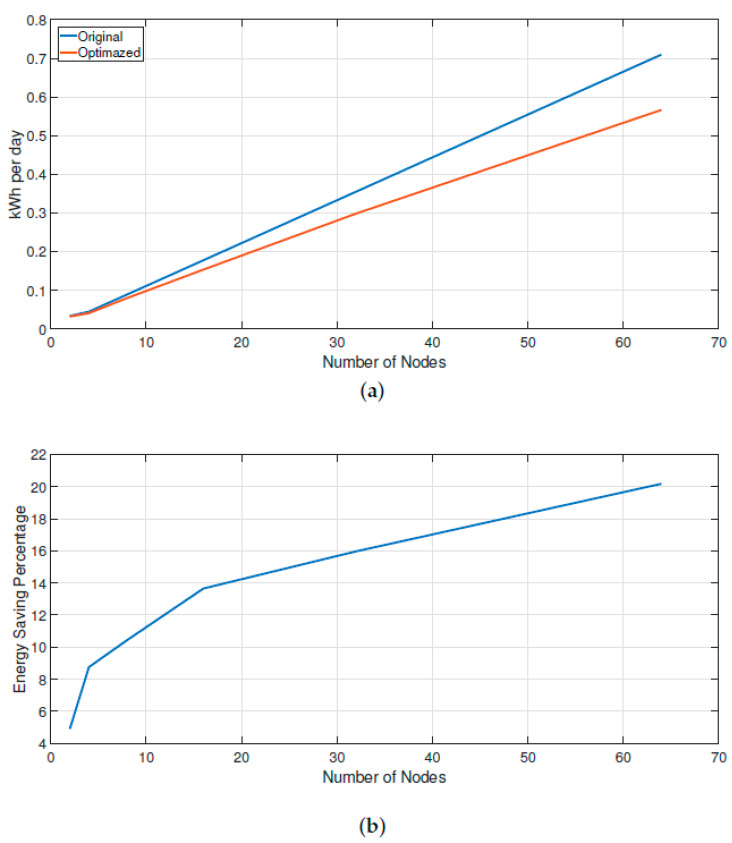
Energy onsumption: (**a**) kilowatt consumption per day; and (**b**) energy saving percentage.

**Table 1 sensors-20-02855-t001:** Example of sensor identifier (SId).

0000 0000	GPS
0000 0001	Passive infrared sensor
0000 0010	Door Control
0000 0011	RFID
0000 0100	Biosensors
0000 0101	Webcam
0000 0110	Temperature Control
0000 0111	Temperature
0000 1000	$CO_{2}$

**Table 2 sensors-20-02855-t002:** ND and QoS impact for β(τ).

Algorithm	β (ND)	β (QoS)
*RDDLA*	*0.0833*	*0.0909*
DETR	1.1667	0.2143
REER	3.000	0.4000
**This proposal**	**0.0417**	**0.0783**

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
