# Peer review of "Energy-Efficient Industrial Internet of Things Software-Defined Network by Means of the Peano Fractal"

_sensors, 2020, doi:10.3390/s20102855_

Round 1

Reviewer 1 Report

The paper presents an algorithm for improving the energy efficiency of IIoT Software-Defined network using the concept of Peano Factal. Despite recognized merits from the proposal, the work should revise substantially to improve the quality and soundness of content.

  1. The work mentioned on software-defined networking technology, but it was clearly described in the proposed system model. It can make hard for readability on the role of this technology in the studied system. I recommend to add a Section (namely, for example System Model) to present the proposed system. 
  2. Some Figures were not described and analyzed in detail, for example Figure 6. I propose to revise the evaluation section to clarify all simulation(experimental) scenarios and corresponding result analysis.
  3. As the work focuses on the energy-efficiency, but it lacks of comparative study with the existing related works. I proposed (if feasible) to conduct this comparative analysis to show the performance of proposal.
  4. Some Figures should be revised to enhance the quality (for example, Figure 4) and match the format of journal.
  5. Since so many abbreviations and notations used in the work, I recommend to add a Table to list all these terms. 
  6. Many long sentences used in the work makes hard to understand. They should be re-phrased for improving readability.

Reviewer 2 Report

This paper brings in a unique and seemingly good idea, namely using Peano fractals for Industrial IoT network deployment and configuration.

While the idea is truly very interesting, the paper requires significant editorial work, language improvement, and ...improved performance evaluation to be digestible and properly supported.

The main technical problems of the paper are

1) missing a proper description of the phases and steps - through pseudo-codes and flowcharts (Ch. 4.2)

2) the given pseudo code is for the Peano fractal algorithm (not cited), and not for the IIoT problem

3) Figure 5 that is supposed to demonstrate the fractal in action seems to miss the point: what is the connection of the fractal and the routers, nodes?

4) the performance evaluation should be improved by describing the RRDLA, DETR, and REER algorithms within the paper to provide a basic understanding of their core idea, and differences to the proposed model.

Other things to improve are the following:

  • The Introduction has several language mistakes and a lot of waaay too long sentences.
  • The Introduction should be split into paragraphs where needed. The list of Chapters should not include their names.
  • Related Work: misses the citation of Xiang-Yang et al (2009).
  • Related Work: should be re-organized, and some further reference added (where Peano Fractals are used for covering planes with radio transmitters). Re-organization:[11] and [12] appears to be the main resources here, discussed through many pages (why?). The authors keep changing the citation back and forth between 11 and 12, which is hard to follow.
  • Before Chapter 4, there is a Beta in line 299. Why? What does it mean?
  • line 335 has some strange capitalization: derivative-free optimization, robustness, flexibility, Easy Administration and Implementation..

An important, final point, again, Figure 5 itself is really good, but it is not described how the nodes are placed and what is the connection (in the Figure) with the fractal.

Round 2

Reviewer 1 Report

Thank you for authors's response. The work was significantly improved. However, the overall work should be reviewed intensively to check grammars, spelling errors by a professional editing service.

Reviewer 2 Report

The authors considered all the issues raised and answered the concerns quite well. The paper is now in good shape; only minimal language corrections are still to be applied.

The Related Works section has really improved, the long sentences are mostly gone. Figure 8 is highly appreciated - now it makes sense! I tried to connect the dots in various ways (one of them was through the increased numbers), but originally failed based on Figure 5. Now it all makes sense with Figure 6 and then Figure 8.

The short descriptions of the compared algorithms are also appreciated.

There are still a few typos left; a further proofread can help to eliminate them. Examples on this minor issue:

<- In 2020 IoT is expected to provide
-> In 2020, IoT is expected to provide

<- According to the above, advances in software for WSN are classified into nine:
-> According to the above, advances in software for WSN can be classified into nine categories:

<- Figure 4. Proposed System Model using a Labelled Transition System[26]. -> Figure 4. Proposed System Model using a Labelled Transition System [26].
